# Tumor Microenvironment, HLA Class I and APM Expression in HPV-Negative Oral Squamous Cell Carcinoma

**DOI:** 10.3390/cancers13040620

**Published:** 2021-02-04

**Authors:** Claudia Wickenhauser, Daniel Bethmann, Matthias Kappler, Alexander Walter Eckert, André Steven, Jürgen Bukur, Bernard Aloysius Fox, Jana Beer, Barbara Seliger

**Affiliations:** 1Institute of Pathology, University Hospital Halle (Saale), 06112 Halle (Saale), Germany; claudia.wickenhauser@uk-halle.de (C.W.); daniel.bethmann@uk-halle.de (D.B.); jana.beer@uk-halle.de (J.B.); 2Department of Oral, Maxillofacial and Plastic Surgery, University Hospital Halle (Saale), 06120 Halle (Saale), Germany; matthias.kappler@uk-halle.de (M.K.); or alexander.eckert@klinikum-nuernberg.de (A.W.E.); 3Department of Oral, Maxillofacial and Plastic Surgery, University Hospital of the Paracelsus Private Medical University of South Nuremberg, 90471 Nuremberg, Germany; 4Institute of Medical Immunology, University Hospital Halle (Saale), 06112 Halle (Saale), Germany; andre.steven@uk-halle.de (A.S.); juergen.bukur@uk-halle.de (J.B.); 5Robert W. Franz Cancer Research Center, Earle A. Chiles Research Institute, Portland, OR 97213, USA; bernard.fox@providence.org; 6Fraunhofer Institute for Cell Therapy and Immunology, 04103 Leipzig, Germany

**Keywords:** oral squamous cell carcinoma, HLA class I, antigen processing machinery, immune cell infiltration, immune escape, prognosis

## Abstract

**Simple Summary:**

Oral squamous cell carcinoma has developed different strategies to escape from T-cell-mediated immune surveillance, which is mediated by changes in the composition of cellular and soluble components of the tumor microenvironment as well as an impaired expression of molecules of the antigen processing machinery leading to a downregulation of HLA class I surface antigens. In depth characterization of these escape mechanisms might help to develop strategies to overcome this tolerance. In this study, human papilloma virus negative oral squamous cell carcinoma lesions were analyzed regarding the protein expression of major components of the HLA class I antigen processing/presentation pathway in correlation to the intra-tumoral immune cell composition, IFN-γ signaling and clinical parameters, which was further confirmed by bioinformatics analyses of datasets obtained from The Cancer Genome Atlas. This novel knowledge could be used for optimizing the design of immunotherapeutic approaches of this disease.

**Abstract:**

Progression of oral squamous cell carcinoma (OSCC) has been associated with an escape of tumor cells from the host immune surveillance due to an increased knowledge of its underlying molecular mechanisms and its modulation by the tumor microenvironment and immune cell repertoire. In this study, the expression of HLA class I (HLA-I) antigens and of components of the antigen processing machinery (APM) was analyzed in 160 pathologically classified human papilloma virus (HPV)-negative OSCC lesions and correlated to the intra-tumoral immune cell response, IFN-γ signaling and to the patient’s outcome. A heterogeneous but predominantly lower constitutive protein expression of HLA-I APM components was found in OSCC sections when compared to non-neoplastic cells. Tumoral HLA-I APM component expression was further categorized into the three major phenotypes HLA-I^high^/APM^high^, HLA-I^low^/APM^low^ and HLA-I^discordant high/low^/APM^high^. In the HLA-I^high^/APM^high^ group, the highest frequency of intra-tumoral CD8^+^ T cells and lowest number of CD8^+^ T cells close to FoxP3^+^ cells were found. Patients within this group presented the most unfavorable survival, which was significantly evident in stage T2 tumors. Despite a correlation with the number of intra-tumoral CD8^+^ T cells, tumoral JAK1 expression as a surrogate marker for IFN-γ signaling was not associated with HLA-I/APM expression. Thus, the presented findings strongly indicate the presence of additional factors involved in the immunomodulatory process of HPV-negative OSCC with a possible tumor-burden-dependent complex network of immune escape mechanisms beyond HLA-I/APM components and T cell infiltration in this tumor entity.

## 1. Introduction

As one of the eight “hallmarks of cancer”, the interaction between immune and tumor cells plays an integral role in controlling the initiation and progression of malignant diseases. A growing body of evidence of changes in the expression of immune modulatory molecules on tumor cells and on cells of the tumor microenvironment (TME) as well as of alterations in the immune contexture, in particular the nature, composition, density, localization and function of immune cell subpopulations, soluble and physical factors, continuously leads to a better understanding of the tumor immune surveillance and immune escape [1]. The characterization of these mechanisms will give insights into the development of innate and acquired resistances to immunotherapies [2,3], which is crucial for improving the efficacy of immunotherapies for the treatment of cancer patients [4,5].

In murine experimental models as well as in human malignant tumors various immune escape strategies resulting in T cell tolerance and loss of T cell recognition have been described. These include (i) lack or downregulation of tumor antigen expression, (ii) loss or reduced expression of HLA-I surface molecules due to impaired expression of APM components, (iii) increased expression of immune suppressive molecules, e.g., the programmed death-like receptor ligand 1 (PD-L1) and the non-classical HLA-G and HLA-E antigens, (iv) downregulation of the interferon (IFN) signal pathways, (v) activation of oncogenic pathways and (vi) alterations in the expression of inflammatory cytokines, metabolites and pH as well as (vii) transcription factors [6,7,8,9]. Despite a loss of heterozygosity and somatic mutations of HLA-I, antigens have been detected in different tumor entities [10,11,12], their frequency is relatively rare suggesting that the impaired HLA-I surface expression of tumors is mainly due to a deregulation of HLA-I APM components, such as the transporter associated with antigen processing (TAP) subunits and low-molecular-weight proteins (LMPs), mediated by epigenetic, transcriptional and/or post-transcriptional control [9,13]. This can occur at each step of the complex HLA-I APM and affect the anti-tumoral T cell responses [14,15,16,17]. Furthermore, an immune suppressive TME mediated by, e.g., cellular and soluble components as well as by the interaction of the intra- and peri-tumoral immune cells negatively interferes with the immune control and could influence the prognosis of tumor patients and their response to (immuno)therapies [14,18].

Despite recent treatment advances, oral squamous cell cancer (OSCC) has caused 700,000 new cases and 380,000 deaths worldwide [19], still one of the leading causes of cancer worldwide with an overall poor prognosis [20,21]. In industrialized countries, it is mainly attributed to smoking and alcohol abuse and is in about 5% of cases associated with human papilloma virus (HPV) infection [22]. Early disease is treated by either surgery or radiotherapy, while for recurrent/metastatic disease, palliative chemotherapy is currently the standard of care [23]. Despite advances in cytotoxic therapies and surgical intervention, the prognosis of patients is still limited with only a little improvement in their outcome over the last decades [24]. Cancer immunotherapies, in particular, immune checkpoint inhibitors (CPI), have been shown to increase the overall survival (OS) of OSCC patients, which led to the FDA approval of the anti-PD-L1 antibody Nivolumab for their treatment [25,26]. Despite these promising results, only transient responses or durable responses to only a limited number of patients are generated, while the majority of OSCC patients progress and die of their disease [27]. Thus, there is an urgent need to identify prognostic and predictive markers and novel therapeutic targets. Emerging evidence of an enriched immune landscape with key immunological features and prognostic relevance in OSCC exists, which was confirmed by in silico analyses of TCGA data [28]. In colorectal cancer (CRC), the quantification of adaptive immune cells composed of CD3^+^ T lymphocytes with a cytotoxic (CD8) and memory (CD45RO) phenotype located at the invasive margin (IM) and tumor center (TC) called “immunoscore” was established as a risk predictor more powerful than TNM and histologic grading in this context. The use of the immunoscore as a consensus biomarker [29,30] and as a tool to distinguish between responders and non-responders to CPI therapies [31] might be also useful for OSCC patients.

Concerning HPV^−^ OSCC, our group recently proposed a more sophisticated immunoscore termed “cumulative suppression index” (CSI). By evaluating the density and geography of the TME, in particular the spatial distance of CD3^+^/CD8^+^ T cells to CD3^+^/CD8^−^/FoxP3^+^ T cells, the CSI was of prognostic relevance in large tumors independent of well-established prognostic factors, and it had a strong effect on the patient’s OS [32]. However, OSCC cells have developed different strategies to counteract immune recognition involving both intrinsic cancer related factors and extrinsic mechanisms, such as an immune suppressive TME and the downregulated expression of HLA-I APM surface antigen components recognized by tumor-specific T cells due to the reduced expression of HLA-I APM components [6]. Mapping these escape mechanisms might help to develop strategies to overcome this tolerance [33]. Therefore, the expression of HLA-I heavy chain (HC), β_2_-m and four distinct APM components (TAP1, TAP2, LMP2 and LMP10) was determined in 160 HPV^−^ OSCC using immunohistochemistry (IHC) and qPCR, respectively. These data were correlated to tumor specific characteristics, the composition of the TME and the patients’ outcome and were further validated by evaluation of corresponding data of The Cancer Genome Atlas (TCGA) from 103 HPV^−^ HNSCC cases.

## 2. Results

### 2.1. Correlation of Clinicopathological Data with the Overall Survival of OSCC Patients

A comprehensive overview of the clinicopathologic data and the OS of the 160 HPV^−^ OSCC patients included in this study is summarized in Table 1. Gender and age had no influence on the survival, while the T stage and the presence of metastases significantly correlated with poor survival. These data were confirmed by TCGA cohort analysis of 103 HPV^−^ HNC, where no prognostic differences concerning gender and age of the patients were reported either. Instead, a better survival was found for HPV^−^ HNC patients at low N stage. Unexpected, but in line with other studies, the WHO tumor grading had no prognostic significance in both collectives (Table 1) [34].

As demonstrated in Table 2, locally extended tumors received radiation therapy, while adjuvant standard chemotherapy was administered in all cases with tumor dissemination irrespective of tumor size, but in particular in patients with a T1/2 tumor.

### 2.2. HLA-I/APM Component Expression in OSCC Tumors and Its Correlation to Tumor Staging

In order to determine the expression levels of HLA-I/APM components in OSCC, protein expression of HLA-I HC, β_2_-m and the APM molecules TAP1, TAP2, LMP2 and LMP10 were analyzed by IHC of formalin-fixed, paraffin-embedded (FFPE) tumor lesions [35]. IHC staining revealed at least slightly reduced but heterogeneous cytoplasmic, membranous and/or nuclear expression levels depending on HLA-I and APM components analyzed when compared to the surrounding non-malignant cells (data not shown).

Based on the HLA-I (HC/β_2_-m) and APM (TAP, LMP) component expression pattern in the FFPE lesions, OSCC were categorized into four distinct HLA/APM phenotypes (phenotype I–IV, see Materials and Methods). In short, while phenotype I expressed high levels of both the HLA-I HC/β_2_-m and APM components (**HLA-I^high^/APM^high^**), phenotype II displayed a concordant low expression pattern of both components (**HLA-I^low^/APM^low^**). Phenotype III expressed discordant levels of HLA-I HC/β_2_-m and high levels of APM components (**HLA-I^discordant high/low^/APM^high^**). Phenotype IV with discordant HLA-I and low APM expression was omitted from further analyses due to the low group size (Table 3). It is noteworthy that the three HLA/APM phenotypes were uniformly distributed over the different T stages (Table 4). As the HC10 antibody only recognizes the HLA-I HC, corresponding fresh frozen OSCC tissues from five FFPE OSCC lesions with phenotype I and five OSCC lesions with phenotype II were stained with the HLA-A/B/C antibody W6/32, recognizing the functional trimeric HLA-I HC/β_2_-m and peptide complex [36]. In these selected cases, the expression levels of HLA-I HC together with β_2_-m staining were comparable to those obtained by employing the W6/32 monoclonal antibody (mAb). These findings suggest expression of the functional trimeric HLA-I complex at least in the OSCC cases analyzed (Figure 1, Appendix A). As a further control, HLA-I HC and β_2_-m mRNA expression was analyzed in a subset of OSCC specimens also demonstrating concordant expression profiles when compared to the protein data (data not shown).

### 2.3. Impact of Tumoral HLA-I/APM Expression on the Intra-Tumoral Immune Cell Cross Talk

To elucidate the impact of OSCC HLA-I/APM expression on the immune cell repertoire, the HLA-I HC and/or β_2_-m expression data were correlated to the frequency and the composition of intra-tumoral T cell subpopulations recently acquired by multispectral imaging within the same sample collective [32] and to the UICC tumor stages. As demonstrated in Figure 2 within all T stages, the density of CD8^+^ T cells was significantly higher in the HLA-I HC/β2-m^high/high^ group than in the two other groups.

In detail, the intensity of tumoral HLA-I HC/β_2_-m surface expression correlated highly significant positively with the density of intra-tumoral CD8^+^ T cells (*p* = 0.008 resp. 0.003), but negatively with the number of CD8^+^ T cells adjacent to FoxP3^+^ cells within a radius of 30 µm (*p* = 0.009 resp. 0.007) (Table 5). Furthermore, a positive correlation between the expression of LMP2 and the density of CD8^+^ T cells was detected (*p* = 0.003). In this context, it is noteworthy that we found a positive correlation of the intra-tumoral JAK1 expression with the intra-tumoral density of CD8^+^ T cells (*p* = 0.024), while no correlation of JAK1 expression with the HLA-I HC, β_2_-m and APM component expression was found (data not shown).

In silico analyses of TCGA data were in line with these findings as the mRNA expression levels of the major HLA-I/APM genes positively correlated with the quantity of the T cell infiltrate demonstrating a significant association of CD8A with the HLA-I HC, β_2_-m, TAP1, TAP2, LMP2 and LMP10 (Table 6, Appendix A).

To further evaluate the probability of the induction of HLA-I/APM expression by T cell secreted IFN-γ, basal and IFN- γ-regulated transcription and protein expression levels in three human HNSCC cell lines were determined using qPCR, Western blot analyses and flow cytometry. In this approach constitutive, but rather low, mRNA levels of HLA-I APM components were detected in all three analyzed cell lines, which was even more pronounced at the protein level and accompanied by low HLA-I surface expression levels. IFN-γ treatment upregulated both HLA-I and APM mRNA and protein expression (Appendix A) suggesting a direct link between HLA-I/APM upregulation and T cell induced IFN-γ secretion.

### 2.4. Impact of OSCC HLA-I/APM Expression on the Outcome of the Patients

To elucidate the clinical impact of the different HLA-I/APM phenotypes of OSCC, the HLA-I APM expression results were set in relation to the OS of the patients using uni- and multivariable Cox proportional hazard regression and Kaplan–Meier curves. Here, an increase in HLA-I HC and β_2_-m expression negatively affected the OS independent of the tumor size. We next analyzed whether this effect was influenced by the T stage or, more specifically, the tumor mass. When considering T1/2 vs. T3/4 tumors, no significant effect was found (Table 7). In contrast, a significantly worse OS and 4.4-fold higher risk of death (*p* = 0.003; multivariable Cox regression, adjusting for UICC N-stage and grading) was detected in phenotype I [HLA-I^high^/APM^high^] in comparison to phenotype III [HLA-I^discordant low/high^/APM^high^] and to a lesser extent to phenotype II [HLA-I^low^/APM^low^] by solely focusing on T2 tumors (Figure 3). Based on these data, we assume that the different results of T1 versus T2 tumors were due to the much better OS of patients with T1 tumors independently of the immune phenotype (Table 1).

In line with these data, in silico analyses employing the XENA database (https://xenabrowser.net/) revealed a reduced OS in HLA-I^high^/APM^high^ HNSCC in comparison to HLA-I^low^/APM^low^ HNSCC lesions. In this study, higher levels of TAP1 expression correlated with a better OS (Table 8).

The expression data of HLA-I and APM components were obtained from the TCGA of 103 HPV^−^ OSCC lesions. The mRNA expression patterns of the APM components TAP1, TAP2 and LMP7 within the phenotypes I and II were correlated to the OS of HPV^−^ HNC patients. The red background in phenotype I indicates a lower OS when TAP1, TAP2 and LMP2 genes and HLA-A, HLA-B and β_2_-m are highly expressed (*p* ≤ 0.1), while the grey background indicates no significant differences in the phenotype I. Low expression levels of TAP2, LMP2 and HLA in phenotype II were associated with a better OS (green background), while the grey background indicates no significant differences (*p* ≤ 0.1).

## 3. Discussion

It has become evident that an altered expression pattern of HLA-I antigens and HLA-I APM components in tumors represents an important immune escape mechanism from CTL-mediated elimination in distinct tumor types and is frequently associated with a poor prognosis and resistance to immunotherapy. These changes have been linked to alterations in the TME [37,38,39,40]. In HNSCC, an immune suppressive equilibrium has been identified, which is characterized by a low frequency of dysfunctional immune effector cells and an altered expression of immune modulatory molecules of HNSCC cells [41]. In HPV^+^ HNSCC, the distinct underlying immune escape mechanisms have been well identified [7,42], while there is still a lack of data concerning immune evasion of HPV^−^ OSCC lesions. Yet a growing body of literature suggested that the immune system plays an important role in the carcinogenesis of HNSCC independent of the HPV status. To get more insight into these processes, this study examined the expression and localization of HLA-I and APM components in HPV^−^ OSCC lesions and correlated the expression pattern with the frequency and density of the intra-tumoral immune cell infiltrate and clinical parameters to identify prognostic markers of OSCC progression and regulatory pathways, which can be potentially useful in the therapeutic setting. Impaired expression of HLA-I APM components is a frequent event in solid tumors when compared to non-malignant cells, thereby progressively inhibiting the ability of CD8^+^ CTL to recognize tumor cells [7,38,43,44,45]. Indeed, in the panel of HPV^−^ OSCC lesions analyzed in this study, we detected an impaired HLA-I APM expression in comparison to adjacent non-tumorigenic tissue, which was more prominent for HLA-I HC and β_2_-m than for the peptide transporters and proteasome subunits. Since this effect significantly varied within our OSCC collective, we classified the tumors into three major phenotypes according to their HLA-I APM component expression (HLA-I^high^/APM^high^, HLA-I^low^/APM^low^, HLA-I^discordant high/low^/APM^high^). The frequency of the three main HLA-I/APM phenotypes ranged from 12% (HLA^low^/APM^low^) to 56 % (HLA-I^discordant high/low^/APM^high^) as shown in Table 3. The reason for the impaired HLA-I/APM expression of phenotype II might be structural alterations or epigenetic silencing of both HLA-I and APM components, while in phenotype III, the APM component expression is functional, and only the HLA-I expression is reduced or discordant, which might be foremost caused by deregulation of HLA-I expression at the transcriptional and posttranscriptional level rather than structural alterations. The molecular mechanisms leading to the distinct HLA-I/APM component expression pattern in the various HNSCC samples requires in depth analysis, such as genomic sequencing and determination of the methylation status of different molecules. However, the results suggest that HNSCC employ distinct immune evasion strategies by affecting multiple components of the HLA class I antigen presentation pathway at different levels.

Interestingly, while the HLA-APM phenotypes were equally distributed over the UICC T stages, they were associated with a distinct TME regarding the type, density and spatial distribution of infiltrating immune cells. As expected, a correlation of HLA-I/APM expression with the immune cell infiltrate demonstrated a direct link of high intra-tumoral CD8^+^ T cell infiltration with high levels of HLA-I APM component and HLA-I surface expression. In detail, the mechanistic link is, at least in part, IFN-γ expression, as shown by us in the positive correlation of the intra-tumoral CD8^+^ T cell density to JAK1 expression, used as a surrogate marker for IFN-γ signaling/inducibility. When looking at results from our HNSCC cell lines, we found constitutive, but rather low, mRNA levels of HLA-I APM components, which could be enhanced upon IFN-γ treatment, as it has been shown previously [5]. This underpins the hypothesis that a deregulation rather than structural alterations of these molecules might be responsible for the altered HLA-I/APM component expression in OSCC tumors. In this context, it is important to consider that the striking difference between the HLA-I/APM phenotypes might be linked to the FOXP3:CD8 ratio, which was more than threefold higher in phenotypes II and III compared to phenotype I.

Thus, our experimental data suggest that the immunological control of phenotype I tumors might be linked with the patients’ outcome. Unexpectedly, the correlation of the HLA-I/APM phenotypes to clinical data revealed that the HLA-I^high^/APM^high^ subgroup, despite a high intra-tumoral CD8^+^ frequency and a rather low FOXP3:CD8^+^ ratio, had an adverse outcome, which was most prevalent in T-stage 2 disease. Yet, most likely due to their overall excellent prognosis, this effect was not detected in T1 tumors. This is in line with a number of studies demonstrating a link of high levels of β_2_-m to a poor prognosis [46,47]. Experimental models also suggest that an increased migration and invasion of OSCC cells is the reason for this survival disadvantage [48]. Assuming that the high levels of HLA-I HC and β_2_-m in the phenotype I OSCC tumors might be often due to the IFN-γ secretion of T cells, an explanation for the adverse patients’ outcome might be caused by a systemic exhaustion leading to dysfunctional T cells both in blood and in the TME as described by others [49]. It is noteworthy that in our study, the functionality of intra-tumoral T cells was not analyzed, since we primarily focused on the HLA-I/APM expression patterns. In contrast to the general dogma, patients with an HLA-I^discordant high/low^ phenotype had a significant better prognosis compared to the HLA-I^high^ phenotype, which is in line with a report in CRC, in which HLA-A2-positive patients had a poorer tumor differentiation [50]. However, in our study this finding was limited to UICC T-stage 2 tumors suggesting that the tumor burden might play a key role in the stage dependent immunological control of OSCC. Last but not least, oxygenization of the TME might help in explaining our results. It was described in OSCC that a hypoxic microenvironment results in a hypoxic gene signature, which negatively interferes with immune surveillance and is associated with a poor clinical outcome. This might be of prognostic relevance in early stage OSCC, as recently demonstrated for colorectal cancer [51,52]. However, the exact underlying molecular mechanisms remains to be identified.

Next to melanoma, non-small cell lung cancer and CRC, in which a relationship between immune cell activity, HLA-I antigens, clinical outcome as well as response to CPI has been proven [18,37,53], our group has shown a significant prognostic value of the presence and quantity of tumor infiltrating lymphocytes (TIL) in OSCC [32]. We propose that beyond the established cumulative suppression index, including the parameters CD8, FOXP3 distance to CD8 and PD-L1 expression, the score would have to be extended at least by the “HLA-I expression”, since a significant adverse outcome in tumors with an HLA-I^high^/APM^high^ phenotype and high intra-tumoral CD8^+^ cell content was found.

In conclusion, we state that a multifactorial tumor protecting microenvironment rather than only a reduced HLA-I/APM expression determines the outcome of OSCC, especially in early T stages of this disease. This hypothesis is based on the fact that neither the intra-tumoral CD8^+^ cell content nor the downregulation of HLA-I alone predict outcome especially in limited tumor disease. These data conclusively emphasize that there might exist a plethora of other soluble and physical factors or immune cells negatively interfering with the OSCC patients’ responses to immunotherapy.

## 4. Materials and Methods

### 4.1. Basic Patient’s Characteristics, Inclusion Criteria and Study Approval

The basal patient’s cohort consists of 246 consecutive untreated OSCC patients who either had the initial biopsy and/or were treated with surgery at the Department of Oral and Maxillofacial Surgery, University Hospital Halle (Saale), Germany, collected between 1995 and 2014. The cohort underwent surgical procedure, and in all cases, the resection of the primary OSCC was done in combination with a neck dissection. Here, ipsilateral to the site of primary tumor levels 1–5 were dissected, contralateral the levels 1 and 2. The resection margins were 1 cm in all dimensions. Dependent on UICC stage, an adjuvant radio-/radiochemotherapy was added.

The study was carried out in compliance with the Helsinki Declaration and was approved by the Ethics Committee of the Medical Faculty of the Martin Luther University Halle-Wittenberg (study #2017-81 and 2020-103). All cases had written consensus of patients. Seventy-two cases were omitted from the study due to the dimension of suitable FFPE tissue or to insufficient tissue quality. All remaining tissue samples were tested for HPV association by either p16 IHC (Zeta Corp., Clone G175-405, dilution 1:25, Arcadia, CA, USA) and/or HPV PCR (Zytovision HPV Chip 1.0, Bremerhaven, Germany). Seventeen patients were omitted from the study because of a positive HPV status resulting in 160 cases for analyses.

With the latest update of the cohort on 1 July 2020, after a median follow-up time of 27 months, 53% of the cohort (85 patients) had deceased with a median survival time at 22 months (mean 16 months). The basic clinical and histopathological data, therapeutic regimens and their association with OS were summarized in Table 1 and Table 2. The tumors were classified as T stage 1 (n = 32; 20%), 2 (49; 31%), 3 (24; 15%) and 4 (55; 34%) with and without nodal or distant metastasis.

### 4.2. Standard and Multiplex Immunohistochemistry

Immunohistochemical analyses of tissue samples were performed as recently described [32]. Briefly, FFPE tissue probes were incubated with the following primary mAbs: NAMB (β_2_-m; SP09-36; dilution 1:50), HC10 (against free HC of HLA-B and -C molecules; HSP09-35; 1:2500), NOB1 (TAP1; SP12-156; 1:50), NOB2 (TAP2; SP12-157; 1:50), SY-1 (LMP2; SP08-118; 1:200), TO-7 (LMP10; SP08-225; 1:200) and 6G4 (JAK1; cell signaling; 1:100). To determine, whether biologically active (β_2_-m-bound) HLA-A/B/C complex was expressed on OSCC, we performed IHC on fresh frozen tumor tissues of 10 exemplary cases was performed using the W6/32 mAb recognizing the HLA-A, -B and -C HC/β_2_-m antigen complex (dilution 1:10) [36]. After fixation with formalin for 3 min followed by a washing step, the specimens were incubated with the primary antibody for 30 min, a post-block reagent for 20 min and a HRP polymer for 30 min all at room temperature with a washing step in between. Visualization was done with DAB and hemalaun as described above.

Immunohistochemistry (IHC) results were semi-quantitatively evaluated utilizing the immune reactive score (IRS) as described by Remmele et al. [35]. In short, for evaluation of membranous, cytoplasmic and nuclear staining intensity (0—negative, 1—low, 2—moderate, 3—strong positive) as well as percentage of stained cells (0—negative, 1—<10%, 2—10–50%, 3—51–80%, 4—>80%) were evaluated and the IRS then calculated as the product of both ranging from 0–12. Ten percent of all cases plus all cases presenting any technical challenges were independently evaluated by two pathologists (CW, DB) and co-reviewed to harmonize and ensure reproducibility of the scoring. One pathologist (DB) scored all the remaining cases.

For correlation analyses of the expression of HLA-I and APM components with the frequency and localization of immune cell subpopulations within the tumor, multispectral imaging data from the same patient cohort (n = 108 with immune cell data) previously published by us were employed [32].

### 4.3. Definition of the HLA-I/APM Phenotypes

Based on the HLA-I (HC/β_2_-m) and APM (TAP, LMP) component expression pattern, OSCC were categorized into four distinct HLA/APM phenotypes. In detail, based on the median IRS of membranous HLA-I HC and β_2_-m expression, HLA expression was dichotomized in “HLA-I high” (median ≥ 4) and “HLA-I low” (median ≤ 3). Based on the median IRS of cytoplasmic TAP1 and TAP2 as well as the median IRS of nuclear LMP2 and LMP10, APM expression was dichotomized in “APM high” (median ≥ 4) and “APM low” (median ≤ 3). As a consequence, four distinct phenotypes (I-IV) could be defined: (i) phenotype I expressed high levels of both the HLA-I HC/β_2_-m and APM components (“**HLA-I^high^/APM^high^**”), (ii) phenotype II displayed a low expression pattern of both HC/β_2_-m and APM components (“**HLA-I^low^/APM^low^**”), (iii) phenotype III expressed discordant levels of HLA-I HC/β_2_-m (HLA^high/low^ or HLA^low/high^) and high levels of APM components (“**HLA-I^discordant high/low^/APM^high^**”) and phenotype IV displayed discordant levels of HLA-I HC/β_2_-m (HLA^high/low^ or HLA^low/high^) and low APM levels (“**HLA-I^discordant high/low^/APM^low^**”).

### 4.4. Bioinformatics

*In silico* analysis was performed using the R2 web tool (http://r2.amc.nl) in order to predict the correlation of HLA-I APM components, CD4, CD8, CD274 and FoxP3 with the mRNA expression of the different genes in HPV^−^ head and neck cancer (HNC) patients. For this, the TCGA dataset “Tumor head and neck squamous cell carcinoma” was chosen (n = 520). HPV^+^ HNC patients and patients with unknown HPV status were omitted from the calculations, while HPV^−^ HNC patients (hpv_status_p16-negative, 103 patients) with data available for the respective markers were analyzed. The 2log expression ratio was compared, and a linear regression was calculated. A *p*-value ˂ 1 × 10^−3^ was considered as significant.

For the OS analyses, the XENA database (https://xenabrowser.net) was employed for the TCGA Head and Neck Cancer study (n = 604). Only primary tumor samples from HPV^−^ patients (hpv_status_p16-negative, 103 patients) were included. A *p*-value ˂ 0.1 was considered as significant.

### 4.5. Statistical Analyses

To evaluate the relationship between HLA-I/APM expression, tumor stage and outcome, two groups (low vs. high protein expression levels) were selected for each marker based on an even distribution of patient numbers in these groups. Cox’s regression hazard model and Kaplan–Meier analyses were used to estimate a correlation of HLA-I and APM expression with OS of OSCC patients. The Cox model was adjusted for prognostic effect of co-variables (T stage, N stage and grading), and relative risk (RR) was calculated. The OS was calculated from the day of histological tumor diagnosis until the date of last follow-up or (if applicable) death. The interrelationships between the different HLA-I antigens and APM components were tested with the Spearman’s rank correlation (r_s_, correlation coefficient). Multiple hypotheses corrections were not applied. A probability (*p*) of <0.05 was defined as a significant result and marked with a star (or if lower than 0.01 with two stars). If not otherwise specified, the results from the cell culture experiments were expressed as mean of at least three biological replicates including standard deviation. All statistical analyses were performed using SPSS software version 20.0 (SPSS Inc., Chicago, IL, USA) and Microsoft Excel 2010 (Microsoft Corporation). For the t-test, two samples assuming unequal variances have been selected.

### 4.6. In Vitro Induction of HLA-I/APM Expression

For determination of the constitutive and IFN-γ inducible HLA-I/APM expression in vitro, the human head and neck squamous cell carcinoma (HNSCC) cell lines SAS, Cal33 and FaDu were cultured in RPMI medium supplemented with 10 % (*v*/*v*) fetal calf serum (PAN, Germany), 2 mM L-glutamine, 100 units/mL penicillin and 100 μg/mL streptomycin at 37 °C in 5% (*v*/*v*) CO_2_ humidified air and were left untreated or treated for 24 or 48 h with recombinant interferon (IFN)-γ (200 µg/mL). Analysis was performed either directly or employing cell pellets, stored in liquid nitrogen until further use.

### 4.7. RNA Isolation, Semi-Quantitative and Quantitative PCR

For evaluation of HLA-I/APM expression by the tumor cell lines as well as by a subcollective of OSCC with availability of FFPE and fresh frozen sections and specimens of normal oral mucosa, total RNA was extracted using the NucleoSpin RNA extraction kit (Macherey-Nagel, Düren, Germany) according to the manufacturer’s instructions and converted to cDNA using the cDNA synthesis kit and oligo-dT primer from Thermo Scientific. Target-specific primers (Appendix A) have been already described [8]. PCR was performed in a Rotorgene and using the Rotorgene 6000^TM^ series software (Qiagen, Hilden, Germany) using GAPDH as control and for normalization.

cDNA was synthesized from double DNase I digested total RNA (500 ng) using RevertAid^TM^ H Minus First Strand cDNA synthesis kit (Fermentas, St. Ingbert, Germany). Thereafter, qRT-PCR was performed on Rotorgene 6000 (Qiagen) with a two-step protocol (40 cycles; 95 °C, 15 s; 60 °C, 30 s) using target-specific primers in combination with Platinum^®^ SYBR^®^ Green qPCR SuperMix-UDG (Invitrogen, Karlsruhe, Germany). Relative mRNA expression levels for specific genes were normalized to that of the housekeeping gene delta-aminolevulinate synthase (ALAS) 1. For determination of the IFN-γ inducibility, the mRNA levels of untreated HNSCC cell lines were set to one, and differences to expression levels of IFN-γ-treated cells were calculated. qPCR analyses were performed on samples obtained from three independent experiments.

### 4.8. Western Blot Analyses

Proteins were extracted from 1 × 10^7^ cells from the three HNSCC cell lines and protein concentration was determined with the Pierce BCA protein assay kit (Fisher Scientific, Schwerte, Germany). For Western blot analyses, 50 μg protein/lane was separated on 8%-12% SDS-PAGEs, transferred onto nitrocellulose membranes (Schleicher and Schüll, Dassel, Germany) and stained with (3%, *w*/*v*) Ponceau S. Immunodetection was performed with specific primary mAbs directed against TAP1, TAP2, LMP2, LMP10, HLA-I HC and β_2_-m (kindly provided by S. Ferrone, Harvard, Boston, MA, USA) as recently described [8]. Staining of blots with a GAPDH mAb (Cell Signaling, Frankfurt, Germany) served as loading control. As secondary antibodies the horseradish peroxidase (HRP)-conjugated anti-rabbit and anti-mouse antibodies (DAKO, Hamburg, Germany), respectively, were used, and the LumiLight WB substrate (Roche Diagnostics, Mannheim, Germany) was employed for detection and visualization with a CCD camera (Eastman Kodak Co., Berlin, Germany).

### 4.9. Flow Cytometry

Flow cytometric analyses were performed on a BD LSRFortessa unit (Becton Dickinson, Biosciences, Heidelberg, Germany). Briefly, 5 × 10^5^ cells left untreated or treated with 200 units/mL of recombinant IFN-γ for 48 h were washed twice in PBS followed by direct staining with the fluorescein isothiocyanate (FITC)-labeled mouse anti-human HLA-I mAb (W6/32) and the respective isotype control (Beckman Coulter, Krefeld, Germany) and then measured on a flow cytometer. For data evaluation, the Kaluza software (Beckman Coulter) was applied. The results are expressed as mean-specific fluorescence intensity (MFI) obtained from at least three independent experiments.

## 5. Conclusions

Immunotherapy is a promising tool for the treatment of different malignancies. However, the survival benefit is limited to a subset of patients. A better understanding of the complex interactions between tumor cells and the tumor microenvironment is urgently needed to improve overall response for cancer patients and improve prognosis towards long-term outcome under immunotherapy.

In this study, the HLA-I and APM component expression was analyzed in a large cohort of OSCC tissues and correlated to the intra-tumoral immune cell infiltrate. These findings were associated to the clinical data of the patients. Unexpectedly, despite a high intra-tumoral CD8^+^ T cell content, tumors with high expression levels of HLA-I and APM components presented a rather unfavorable outcome, which was most significant in smaller (T1/T2) tumors. We conclude that an effective immunomodulatory treatment strategy in OSCC has to include the blockade of more than one immune regulatory axis.

## Figures and Tables

**Figure 1 cancers-13-00620-f001:**
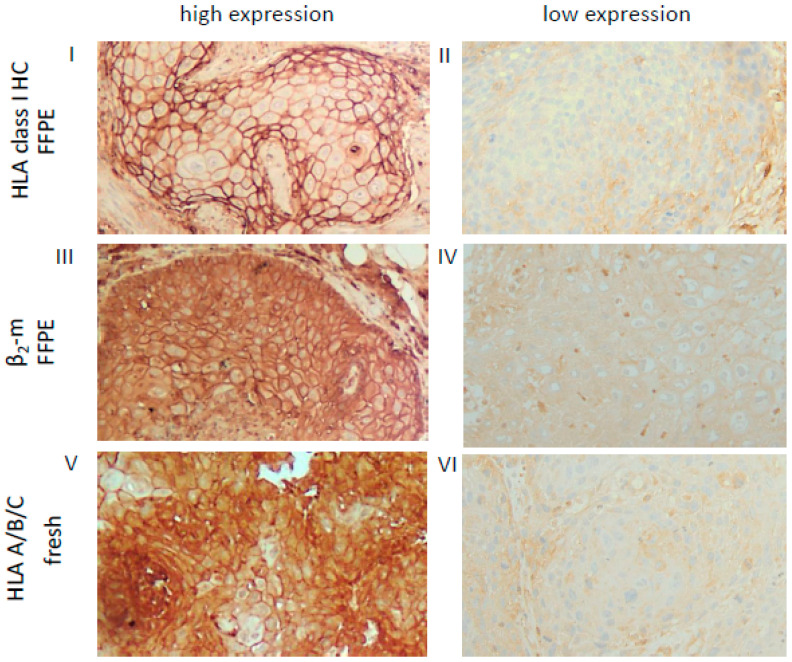
Representative high and low staining patterns of OSCC lesions for HLA-I HC, β_2_-m and the trimeric HLA-I complex. Immunohistochemical analyses using antibodies directed against HLA class I HC (FFPE, **I**, **II**), β_2_-m (FFPE **III**, **IV**) and the trimeric HLA class I A/B/C complex (W6/32; fresh frozen, **V**, **VI**). Representative cases with a high (**I**, **III**) or low (**II**, **IV**) cytoplasmic and membranous expression of HLA class I HC and β_2_-m in FFPE samples are shown and compared to a staining of the trimeric HLA class I complex (**V**, **VI**) on fresh frozen tissues. Each image was done at a 200× resolution.

**Figure 2 cancers-13-00620-f002:**
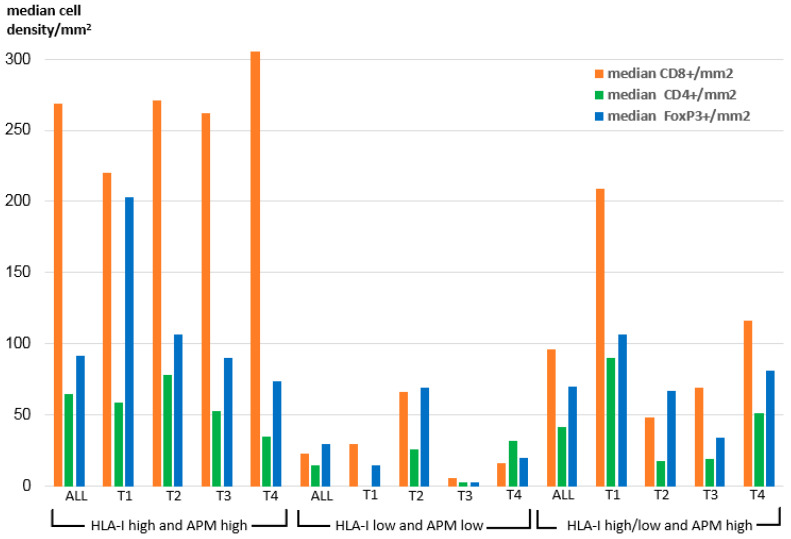
Intra-tumoral median density of T cell subpopulations across the HLA-I/APM phenotypes I-III in correlation to the T stage.

**Figure 3 cancers-13-00620-f003:**
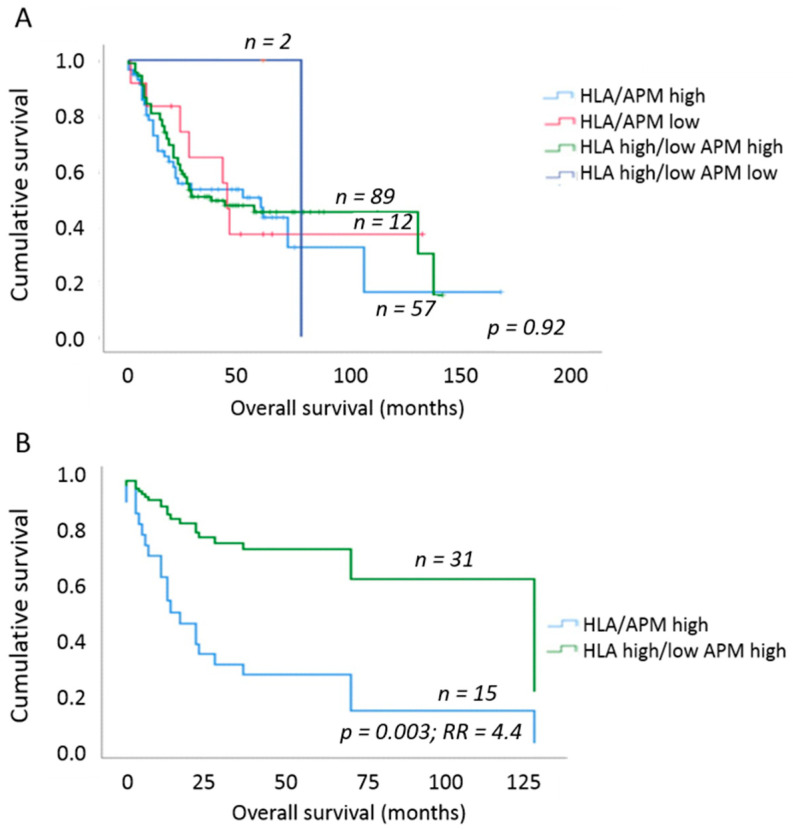
Overall survival of OSCC patients depending on HLA-I/APM phenotype. (**A**) Comparison of OS between the four HLA-I/APM phenotypes irrespective of the tumor size. No clear separation of the survival curves was found. (**B**) OS of OSCC patients when only T2 stage tumors were analyzed. Within this group, patients with an HLA-I^high^/APM^high^ phenotype had a 4.4-fold higher relative risk of death due to OSCC when compared to patients with HLA-I^discordant high/low^/APM^high^ phenotype. Multivariable Cox regression adjusting for UICC N stage as well as grading.

**Table 1 cancers-13-00620-t001:** Clinicopathologic Data and Overall Survival of 160 HPV^−^ OSCC Patients.

Category	No. of Cases	Overall Survival (OS): All Tumors	
		Univariable Analysis	Multivariable Analysis *
		RR; *p*-Value	RR; *p*-Value
*gender*			
men	124		
women	36	0.79; *p* = 0.4	0.87; *p* = 0.64
*age (years)*			
≤60	91		
>60	69	1.17; *p* = 0.49	1.29; *p* = 0.26
*T stage*			
I	32		
II	49	2.54; *p* = 0.025	2.75; *p* = 0.02
III	24	3.04; *p* = 0.018	3.10; *p* = 0.028
IV	55	6.99; *p* < 0.0001	6.92; *p* < 0.0001
*N stage*			
N0	71		
N1–3	89	2.22; *p* = 0.001	1.26; *p* = 0.4
*M stage*			
M0	154		
M1	6	2.41; *p* = 0.057	3.16; *p* = 0.017
*grading*			
1	18		
2	107	0.96; *p* = 0.92	0.65; *p* = 0.24
3	35	0.85; *p* = 0.69	0.58; *p* = 0.2

HPV^−^ OSCC, human papilloma virus-negative oral squamous cell carcinoma; * Cox proportional hazards regression: adjusted for T, N stage and grading; RR, relative risk. Statistically significant data is marked in bold. The frequency of radiation therapy (%) including the median dosage as well as the frequency of standard chemotherapy (%) of patients in different T stages is given.

**Table 2 cancers-13-00620-t002:** Frequency and Intensity of Adjuvant Therapy.

UICC T Stage	Number of Cases	Adjuvant Radiation Therapy Received	Median Dosage (gy)	Adjuvant Standard Chemotherapy Received
I	32	14 (44%)	50.2	24 (75%)
II	49	26 (53%)	50.0	12 (25%)
III	24	17 (71%)	64.0	9 (38%)
IV	55	40 (73%)	64.0	31 (56%)

**Table 3 cancers-13-00620-t003:** Classification of OSCC Lesions According to the HLA-I/APM Expression.

HLA-I/APM Phenotype	Patients (n = 160)	HC	β_2_-m	TAP1	TAP2	LMP2	LMP10	Expression Pattern
I [HLA-I^high^–APM^high^]	57 (36%)	+	+	+	+	+	+	concordant HLA-I/APM expression
II [HLA-I^low^–APM^low^]	12 (8%)	−	−	−	−	−	−	
III [HLA-I^discordant high/low^–APM^high^]	89 (56%)	+	−	+	+	+	+	discordant HLA-I/APM expression
		−	+					
IV [HLA-I^discordant high/low^–APM^low^] *	2 (<1%)	+	−	−	−	−	−	
		−	+					

* omitted from further analyses due to limited group size. A discordant HLA-I HC and β_2_-m expression is marked as ^high/low^.

**Table 4 cancers-13-00620-t004:** Distribution of T Stages Across HLA-I/APM Phenotypes.

HLA-I/APM Phenotype	T1	T2	T3	T4	all Ts
I [HLA-I^high^/ APM^high^]	16 (50%)	15 (31%)	12 (50%)	14 (26%)	57 (36%)
II [HLA-I^low^/ APM^low^]	3 (9%)	2 (4%)	3 (13%)	4 (7%)	12 (8%)
III [HLA-I^discordant high/low^/APM^high^]	12 (38%)	31 (63%)	9 (37%)	37 (67%)	89 (55%)
IV [HLA-I^discordant high/low^/APM^low^] *	1 (3%)	1 (2%)	-	-	2 (1%)

* omitted from further analyses due to limited group size. A discordant HLA-I HC and β_2_-m expression is marked as ^high/low^.

**Table 5 cancers-13-00620-t005:** Bivariable Correlation of the Expression of the HLA-I HC, β_2_-m, APM Components and JAK1 with the Intra-Tumoral T Cell Infiltrate.

Intra-Tumoral T Cell Infiltrate		HLA-I HC	β_2_-m	TAP1	TAP2	LMP2	LMP10	JAK1
	N	r_s_ (*p*-Value)	r_s_ (*p*-Value)	r_s_ (*p*-Value)	r_s_ (*p*-Value)	r_s_ (*p*-Value)	r_s_ (*p*-Value)	r_s_ (*p*-Value)
CD4^+^/mm^2^	108	0.150 (0.12)	0.216 (0.03) *	−0.142 (0.14)	0.168 (0.08) *	0.229 (0.017) *	0.120 (0.22)	0.027 (0.78)
CD8^+^/mm^2^	119	**0.242 (0.008) ****	**0.272 (0.003) ****	−0.172 (0.08) *	0.118 (0.20)	**0.270 (0.003) ****	0.095 (0.30)	**0.178 (0.06) ***
FoxP3^+^/mm^2^	119	0.190 (0.04) *	**0.246 (0.007) ****	−0.239 (0.01) *	0.198 (0.03) *	0.146 (0.11)	0.012 (0.90)	0.115 (0.24)
CD8^+^ to FoxP3^+^ within 30 µm	119	**−0.238 (0.009) ****	**−0.248 (0.007) ****	−0.034 (0.73)	−0.043 (0.64)	−0.223 (0.015) *	−0.024 (0.79)	−0.097 (0.318)

Bold format—statistically significant result, * denotes *p* < 0.05; ** denotes *p* < 0.01.

**Table 6 cancers-13-00620-t006:** Correlation of the mRNA Expression of HLA-I/APM Components with T Cell Markers Using TCGA Data from HPV^−^ HNC Patients.

	HLA-A	β_2_-m	TAP1	TAP2	LMP2	LMP10
**CD4**	r_s_ = 0.351 (*p* = 2.3 × 10^−3^)	r_s_ = 0.458 (*p* = 4.6 × 10^−5^)	r_s_ = 0.394 (*p* = 5.6 × 10^−4^)	r_s_ = 0.372 (*p* = 1.2 × 10^−3^)	r_s_ = 0.328 (*p* = 4.6 × 10^−3^)	r_s_ = 0.474 (*p* = 2.3 × 10^−5^)
**CD8A**	r_s_ = 0.506 (*p* = 5.0 × 10^−6^)	r_s_ = 0.698 (*p* = 7.0 × 10^−12^)	r_s_ = 0.699 (*p* = 1.5 × 10^−11^)	r_s_ = 0.642 (*p* = 9.4 × 10^−10^)	r_s_ = 0.653 (*p* = 3.7 × 10^−10^)	r_s_ = 0.671 (*p* = 7.9 × 10^−11^)
**FoxP3**	r_s_ = 0.347 (*p* = 2.7 × 10^−3^)	r_s_ = 0.460 (*p* = 4.2 × 10^−5^)	r_s_ = 0.445 (*p* = 7.9 × 10^−5^)	r_s_ = 0.45 (*p* = 6.4 × 10^−5^)	r_s_ = 0.342 (*p* = 3.1 × 10^−3^)	r_s_ = 0.392 (*p* = 6.0 × 10^−4^)

TCGA, the cancer genome atlas; Spearman’s Rho test: r_s_—correlation coefficient; statistically significant data are marked in bold. In silico analyses of tumor samples from 103 HPV^−^ HNC patients obtained from the TCGA data bank presented as r_s_.

**Table 7 cancers-13-00620-t007:** Expression of HLA-I/APM components and correlation with T1/2 and T3/4 stage as well as overall survival.

	No. of Cases	Overall Survival (OS)					
		All Tumors		T1–2 Stage Tumors		T3–4 Stage Tumors	
		Univariable Analysis	Multivariable Analysis *	No. of Cases	Multivariable Analysis *	No. of Cases	Multivariable Analysis *
		RR; *p*-Value	RR; *p*-Value		RR; *p*-Value		RR; *p*-Value
*HLA-APM phenotype* *cytoplasmic and membranous* I [HLA-I^high^-APM^high^] II [HLA-I^low^-APM^low^] III [HLA-I^high/low^-APM^high^] IV [HLA-I^high/low^-APM^low^]	57 12 89 2	I vs. III: 0.59; *p* = 0.63	I vs. III: 1.29; *p* = 0.55	31 5 43 2	I vs.III: 1.86; *p* = 0.11	26 7 46 0	I vs. III: 1.08; *p* = 0.82
*HLA-I HC*							
*cytoplasmic*							
IRS 0–3	64			34		30	
IRS 4–12	96	1.53; *p* = 0.07	1.52; *p* = 0.08	51	1.86; *p* = 0.11	49	1.25; *p* = 0.47
*membranous*							
IRS 0–4	100			51		49	
IRS 6–12	60	1.13; *p* = 0.59	1.04; *p* = 0.86	30	1.85; *p* = 0.10	30	0.72; *p* = 0.26
*β_2_-m*							
*cytoplasmic*							
IRS 0–3	48			27		21	
IRS 4–12	112	**1.80; *p* = 0.026**	1.66; *p* = 0.056	54	2.29; *p* = 0.061	58	1.41; *p* = 0.31
*membranous*							
IRS 0–4	105			56		53	
IRS 6–12	55	0.97; *p* = 0.9	0.93; *p* = 0.75	25	1.53; *p* = 0.28	26	0.8; *p* = 0.49
*TAP1*							
*cytoplasmic*							
IRS 0–2	18			8		10	
IRS 3–12	49	0.57; *p* = 0.09	0.57; *p* = 0.13	25	0.89; *p* = 0.87	24	0.56; *p* = 0.20
*TAP2*							
*cytoplasmic*							
IRS 0–4	49			26		23	
IRS 6–12	111	1.31; *p* = 0.27	1.42; *p* = 0.17	55	1.02; *p* = 0.96	56	**2.35; *p* = 0.02**
*LMP2*							
*cytoplasmic*							
IRS 0–4	105			53		52	
IRS 6–12	55	0.99; *p* = 0.96	0.99; *p* = 0.99	28	1.02; *p* = 0.96	27	1.06; *p* = 0.84
*nuclear*							
IRS 0–4	69			34		35	
IRS 6–12	91	1.52; *p* = 0.07	1.56; *p* = 0.054	47	**2.4; *p* = 0.04**	44	1.44; *p* = 0.2
*LMP10*							
*cytoplasmic*							
IRS 0–2	32			22		10	
IRS 3–12	128	1.54; *p* = 0.15	1.27; *p* = 0.43	59	1.34; *p* = 0.5	69	1.33; *p* = 0.51
*nuclear*							
IRS 0–4	21			12		9	
IRS 6–12	139	1.63; *p* = 0.18	1.57; *p* = 0.21	69	1.26; *p* = 0.65	70	2.1; *p* = 0.16

* Cox proportional hazards regression, adjusted for T, N stage and grading; RR, relative risk. Statistically significant data are marked in bold.

**Table 8 cancers-13-00620-t008:** Correlation of OS to the HLA-I/APM component expression using TCGA data of 103 HPV^−^ HNC patients.

**High HLA-A, HLA-B or β_2_-m** **(Phenotype I)**	**TAP1**	**TAP2**	**LMP2**	**LMP10**
**HLA-A**	*p* = 0.028	*p* = 0.04	*p* = 0.068	*p* = 0.131
**HLA-B**	*p* = 0.479	*p* = 0.195	*p* = 0.045	*p* = 0.077
**β_2_-m**	*p* = 0.019	*p* = 0.067	*p* = 0.074	*p* = 0.145
**low HLA-A, HLA-B or β_2_-m** **(phenotype II)**	**TAP1**	**TAP2**	**LMP2**	**LMP10**
**HLA-A**	*p* = 0.679	*p* = 0.04	*p* = 0.059	*p* = 0.557
**HLA-B**	*p* = 0.884	*p* = 0.094	*p* = 0.028	*p* = 0.624
**β_2_-m**	*p* = 0.873	*p* = 0.058	*p* = 0.055	*p* = 0.801

Interrelationship of the HLA-I/APM component expression. Green/red color shade in general symbolizes a statistically significant correlation (*p* < 0.05), whereas green color shade shows a positive- and red color shade shows a negative correlation.

## Data Availability

Not applicable.

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
