# Peer review of "Tumor Microenvironment, HLA Class I and APM Expression in HPV-Negative Oral Squamous Cell Carcinoma"

_cancers, 2021, doi:10.3390/cancers13040620_

Round 1

Reviewer 1 Report

The text has been modified according with the proposed suggestions . The title is now shorter and clearer.

The paper is now suitable for publication.

Reviewer 2 Report

The authors have sufficiently addressed my major concerns. 

This manuscript is a resubmission of an earlier submission. The following is a list of the peer review reports and author responses from that submission.

Round 1

Reviewer 1 Report

The text is very hard to read. The sentences are too long and often the thread of the conversation is lost. Also the title is “dispersive”. The short running title is clearer and better explains the content of the paper. The results are not clearly explained in the text and are reported before the materials and methods. Discussion and conclusions are independent of the results.

Reviewer 2 Report

This clinical study by Wickenhauser C et al showed that high expression levels of HLA-I and APM were correlated with increased infiltration of intra-humoral CD8+ T-cells. However, this group suffered from the worst overall survival. The results are intriguing with a significant indication for clinical biomarker selection. The study is well conceived and well presented. Some minor areas that further enhance the rigor of this study are as follows:

It is well documented, including data presented in this study, that the levels of HLA-I and APM are driven by interferon signaling. This study used cell lines to show interferon treatment up-regulates HLA-I and APM. Although consistent with previous studies, it does not directly correlate with the patient cohort in this study. The IHC staining of IFN-gamma or IFN-beta has been problematic, however, MX1 is a sensitive downstream marker for interferon signaling. It would corroborate the finding by examining MX1 and performing the correlation studies as reported in this manuscript. Overall, this is an excellent study that should be a valuable addition to the literature.

Reviewer 3 Report

In the manuscript by Wickenhauser et. al., the authors analyze MHC1 expression and components of the processing machinery by immunohistochemistry and as part of publicly available datasets. The authors find interesting associations between stage, APC/MHC expression and prognosis. The manuscript is generally well done and needs to be improved with a few more details. 

Major comments. 

1) Can the authors confirm that multiple hypothesis corrections were applied or where they were applied to statistical analysis (bonferoni corrections, etc.), this is unclear at several points in the current text. 

2) The discussion could use expansion on the potential involvement of checkpoint proteins or other immune supressive events (e.g. oxidation states) to help account for the unexpected relationships observed in the results.

3) Can the authors provide greater depth of analysis on the MHC staining patterns and breakdowns of staining patterns in the results. The greater depth may provide pivotal insights into the mechanism of action.  

Minor

  • Line 135-define acronyms (HC=heavy chain)
  • Line 156-“specimens”
  • Tapasin is mentioned on the abbreviations list, but I can’t find a mention in the main paper.
  • If there’s space, a graphical representation of table 3B would be nice.
  • Table 4A/Line 219: These p values don’t match. Where does p=0.003 come from?
  • Figure 2: hard to read legends, better resolution and bigger font would be appreciated.
  • In a 2005 paper, this group described an association between APM expression and HNSCC prognosis. Were they able to replicate this association in this dataset?